# Training intervention to improve hygiene practices in Islamic boarding school in Yogyakarta, Indonesia: A mixed-method study

Vita Widyasari [1,2,¤a]*, Yayi Suryo Prabandari[3], Adi Utarini[4,¤b]*

1 International Health, Public Health Graduate Program, Faculty of Medicine, Public Health and Nursing, Universitas Gadjah Mada, Yogyakarta, Indonesia, 2 Cluster of Public Health Science, Faculty of Medicine, Universitas Islam Indonesia, Yogyakarta, Indonesia, 3 Department of Health Behavior, Environment, and Social Medicine, Faculty of Medicine, Public Health, and Nursing, Universitas Gadjah Mada, Yogyakarta, Indonesia, 4 Department of Health Policy and Management, Faculty of Medicine, Public Health and Nursing, Universitas Gadjah Mada, Yogyakarta, Indonesia

☯ These authors contributed equally to this work.
¤a Current address: Cluster of Public Health Science, Faculty of Medicine, Universitas Islam Indonesia, Yogyakarta, Indonesia
¤b Current address: Department of Health Policy and Management, Faculty of Medicine, Public Health and Nursing, Universitas Gadjah Mada, Yogyakarta, Indonesia
* vita.widyasari@uii.ac.id (VW); adiutarini@ugm.ac.id (AU)

**Data Availability Statement:** All Hygiene training files are available from the figshare database (DOI: 10.6084/m9.figshare.9891200)

## Abstract

### Background

The primary objective of this study was to determine the effect of a training intervention in overall improvement in students' *(santris)* knowledge, behavior, and outcome.

### Methods

A mixed-methods exploratory sequential design was applied. First, qualitative data were collected from three focus group discussions with 20 supervisors and one in-depth interview with school principal to explore current hygiene practices. The information was then used to develop training intervention using either video, poster, and leaflet. To measure the effect, a stepped wedge cluster design with pre- and post-test analyses was conducted. A total of 452 junior high school santris in one Islamic boarding school were non-randomly allocated to either three intervention groups. Outcome measures were knowledge, personal behavior, and room hygiene. Codes and categories were produced in the qualitative analysis, while paired t-tests and Wilcoxon rank tests test were used in the quantitative analysis.

### Results

The qualitative study identified poor practices on personal and room hygiene among the santris and proposed a training intervention. Overall, there was a significant increase in knowledge and personal behavior after the intervention (7.22 ± 1.34 pre-intervention to 7.70 ± 0.74 post-intervention and 9.75 ± 2.98 pre-intervention to 12.16 ± 2.12 post-intervention, respectively, p < 0.001). Room hygiene was significantly improved among boys and those who received leaflets.

**Funding:** The author(s) received no specific funding for this work.

**Competing interests:** The authors have declared that no competing interests exist.

**Abbreviations:** MDGs, Millennium Development Goals; SDGs, Sustainable Development Goals; FGDs, Focus Group Discussions; WASH, Water, Sanitation and Hygiene.

## Conclusion

Having developed a specific training materials, school-based hygiene training intervention improved knowledge and personal behavior. Its effect on room hygiene particularly for female santris needs further strengthening of the intervention in this Islamic boarding school setting.

## Introduction

Hygiene is a basic need of every human being. Despite great achievements during the millennium development goals (MDGs) era, more efforts must be undertaken and sustained to improve hygiene practices. Water, sanitation, and hygiene are still considered important priorities globally, and these basic human needs are clearly stated in one of the targets of the sustainable development goals (SDGs). Goal 6 articulates the aims of achieving access to adequate and equitable sanitation and hygiene for all and putting an end to open defecation, paying special attention to the needs of women, girls, and those in vulnerable situations. Further, poor sanitation and hygiene has significant impacts not only on health but also on safety, well-being, and the educational prospects of women [1].

Despite substantial recent progress in access to a clean water supply and sanitation services in Indonesia, MDGs targets have not been achieved. Indonesian Basic Health Research 2015 survey stated that only 66.8% had access to clean water and only 58.9% had access to basic sanitation (compared to 68.9% and 62.4%, respectively) [2, 3]. Furthermore, several groups need special attention regarding hygiene and sanitation, such as people in rural areas [3], those with special needs [4], and those in specific congregate communities, including orphanages and boarding schools.

Indonesia is a Muslim majority country; therefore, Islamic Boarding Schools play an important role in the educational system. Schools vary in terms of curriculum and facilities [5]. Most students (santris) who live in these boarding schools intend to learn about their religion in a more intensive atmosphere while also pursuing general education. Santris commonly attend boarding schools starting in elementary school.

Islam strongly emphasizes hygiene, and it is stated that cleanliness is half of the faith. Despite efforts to maintain hygiene and sanitation in boarding school settings, many challenges remain. Based on the preliminary study that was conducted by first author in January 2016 at Mawar Islamic Boarding School [6], we found that hygiene facilities and personal hygiene were generally poor, such as shown by poor garbage system, santri's behavior of rarely changed their clothes and eat by hand without washing hands, and so forth. With 10–12 students in one room, low awareness about maintaining room hygiene was also reported, particularly in the girls' room who have much more stuff. The problem persists and the school's efforts to overcome this problem has so far been unsuccessful. In a 2014–2015 Mawar Islamic Boarding School health report [6], 70% of santris in a boarding school were diagnosed with infectious diseases, predominantly respiratory and skin infections. Low awareness about maintaining personal and room hygiene was also reported. Factors that influence adolescent behavior are role models, teachers, supervisors, and peer relationships [7–9]. Many studies on the negative health impacts of poor hygiene and sanitation have been conducted. The effects of these behaviors include a high risk of infections, including scabies [10–13] and genital infection [14].

This mixed-methods study describes the impact of efforts to improve hygiene practices among santris in an Islamic boarding school in Yogyakarta, Indonesia using a three-phase exploratory sequential study design. The purpose of the study was to measure the effect of different interventions on improving hygiene practices among Islamic boarding school students (santris). Our specific objectives were to:

1. Explore santris' personal and room hygiene bevhaiors as well as supervisors' perceptions and proposed solutions to improve hygiene practices;

2. Develop health educational interventions to improve knowledge and behavior on hygiene practices;

3. Describe the santris' reactions toward the interventions; and

4. Measure the differences in knowledge, behavior and outcome before and after the interventions among different interventions and according to gender.

## Materials and methods

### Study site

This study was conducted in one of the biggest modern Islamic boarding schools located in Yogyakarta Special Province, Indonesia, which serves approximately 3,500 santris from the kindergarten to the undergraduate level. The majority of santris (roughly 1,500) are in junior high school. The boarding school occupies a 2,000 $m^2$ area and includes three building blocks with 115 rooms for female santris and three blocks with 90 rooms for males. The rooms are approximately 18 $m^2$ and accommodate 12 children. The rooms are equipped with a bunk bed and small wardrobes. Each block also includes two to three rooms for supervisors. Every supervisor is responsible for attending to three or four santri rooms. This study was conducted in July 2016 until September 2016.

### Study design

This research used a mixed-method study in order to ensure an appropriate design is used in this congregate setting, followed by measuring the effect of intervention. Applying a three-phase exploratory sequential mixed-methods [15], a qualitative study was first understaken to explore perceptions and current hygiene practices among the supervisors and to identify the potential interventions. Based on qualitative findings, the interventions were developed implemented in the following quantitative study. A quasi-experimental study was then conducted in quantitative phase, applying the stepped wedge cluster pre-post design. The cluster was defined as a block of adjacent rooms. The number of rooms per cluster was 8 rooms for girls and 6–7 rooms for boys. Total rooms number in our intervention was 24 girls room and 20 boys room. One room was occupied by 10–12 santris. This design was selected to implement a step-by-step intervention for the entire sample due to limitations in conducting the intervention all at once. First, all study participants were considered to be in the control group, and then the intervention was applied in a stepwise fashion until all participants had received it [16]. The participants were junior high school santris currently enrolled in the program. The subjects were nonrandomly assigned to one of three different interventions (video, poster, or leaflets).

### Data collection and analysis

**Qualitative study.** The participants in the qualitative study were twenty santri supervisors and one school principal. Data were collected through three focus group discussions (FGDs)

with two groups of female supervisors and one group of male supervisors, as well as an in-depth interview with the school principal. In this boarding school, supervisor is a senior student chosen by the school to take the role of supervisor. This designated santri supervisors (male and female) have at least completed senior high school from this school and they are chosen based their high performance for the standard on religious practice and competence. Once they become supervisors, they have to live in the boarding school and are responsible to take care of 30–50 santris', teach the Qur'an while undertaking their own activity, such as working as school staff or studying at university level.

The FGDs covered the following topics: current problems and expectations, approaches and efforts made, and problems in implementation and its effectiveness. The in-depth interview explored existing school hygiene regulations with the goal of data triangulation. The interview and FGDs were between 32 and 53 minutes long, and audio recorded. Transcripts were produced and returned to the participants prior to data analysis as effort of trustworthiness. No further comments were provided by the interviewee or FGD participants following data transcription.

Transcripts were analyzed using OpenCode 4.02 software, and code categories were generated by one coder (VW). The second and third authors (YSP and AU) analyzed one transcript each, and different codes were discussed to obtain consensus. After identifying the codes, we grouped similar codes into categories and core themes were described [12]. The categories were then discussed and agreed upon among all authors.

**Quantitative study.** All new students at the junior high school level were recruited as participants, resulting in a total of 452 santris (Fig 1). Participants were nonrandomly allocated to the intervention due to boarding school conditions. The intervention was conducted at the cluster level, that consists of block of adjacent rooms, separated for female and male students. Students living in a given cluster were allocated in a non-randomized fashion into the groups of intervention. The A to H girls room and A to F boys room received the video intervention, the I to P girls room and G to M boys room received the poster intervention, and the Q to X girls room and N to T boys room were allocated for the leaflet intervention. All rooms were located in the first floor. Girls and boys are located in a separate building and their activities are also separate.

Knowledge was measured before the start of the training intervention and one-week post intervention, while personal and room hygiene was measured by supervisors using questionnaires. Assessments of the santris' personal and room hygiene practices were carried out by supervisors prior to the training intervention and continued on a weekly basis until one month after the last intervention (week 8).

Four instruments were designed following the Kirkpatrick training evaluation method [17]. First, the reaction to the training was measured using a Likert scale ranging from 1 to 5, with 5 being the best value. The questionnaire consisted of five questions about the students' perspectives about the training, adapted from Khasawneh and Al-Zawahreh [18]. The answers to other questions assessing knowledge, behavior, and study outcomes were dichotomous (yes/no). The knowledge questions consisted of eight items from Patrick [19] and Miko *et al.* [20]. The items reflected basic knowledge on personal and room hygiene. Assessment of behavior was based on a sixteen item index of daily hygienic practices adapted from Saffari *et al.* [21]. Finally, the outcome of room hygiene was measured using a ten item index based on the santri's room characteristics.

Data analysis was conducted using Stata 12 software. The analysis was carried out for each four variables (reaction evaluation, knowledge, behavior, and outcome), following the four levels in the Kirkpatrick's training evaluation method [17]. The first level of evaluation was to assess reactions using a Likert scale. The data were presented in a two-by-two table for each indicator and described descriptively using means and standard deviations. The second level

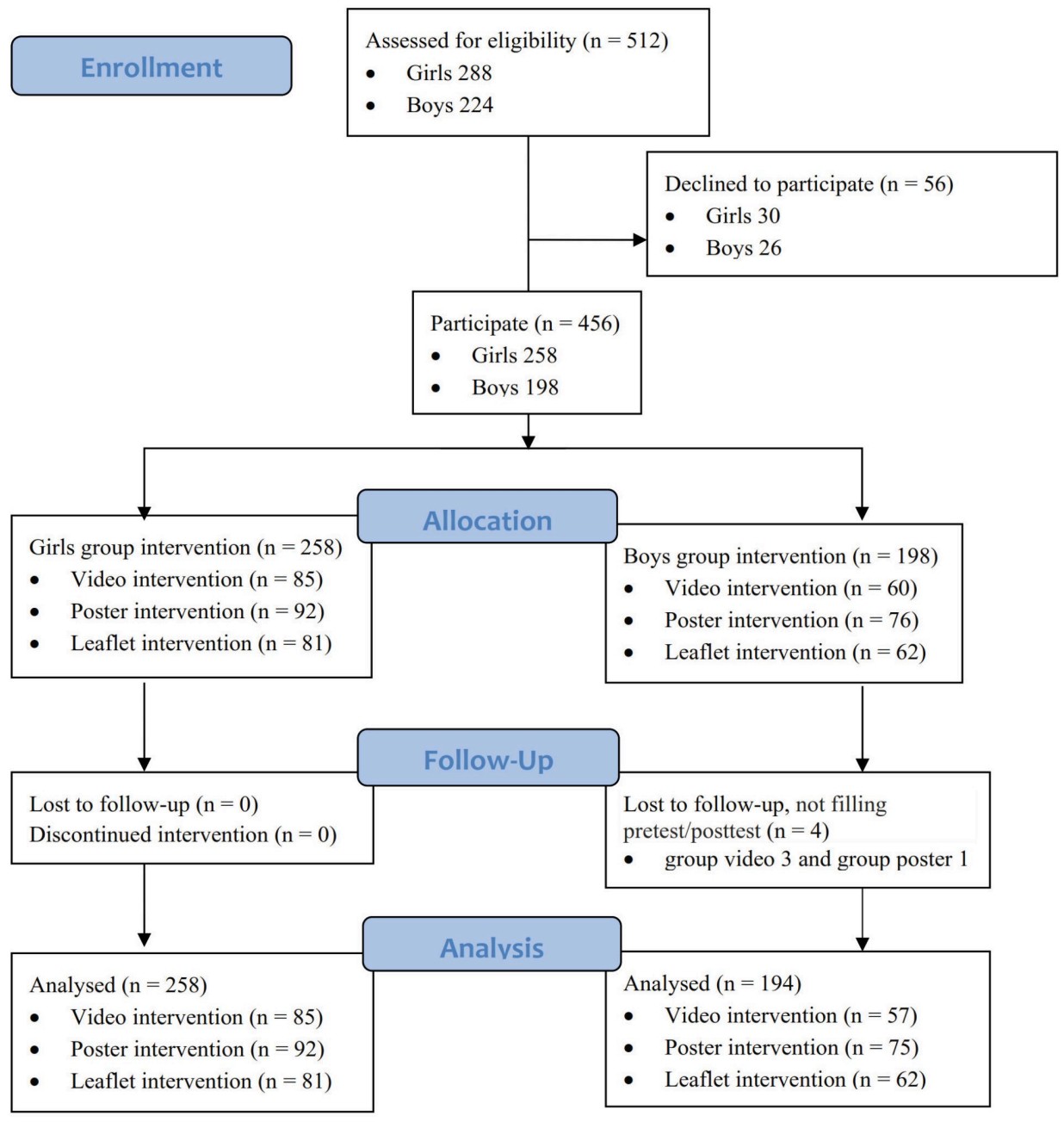

**Fig 1. CONSORT diagram participant allocation.**

of evaluation was to assess knowledge, using paired t-tests to calculate the mean differences between pre- and post-training intervention within the group. Differences between gender were tested using an independent t-test, while differences between the three interventions (video, poster, and leaflet) were tested using one-way ANOVA, continued by Bonferroni post hoc analysis. The third level of evaluation focused on individual behavior, i.e. assessing differences in personal hygiene over the course of the intervention, which was tested using a paired t-test. Gender differences were tested using an independent t-test and similar to the second level of evaluation, differences in hygiene between the three interventions were assessed by

applying a one-way ANOVA, continued by Bonferroni post hoc analysis. Finally, the fourth level was evaluated by examining room hygiene as the effect. For the analysis on room hygiene (outcome), it did not meet the criteria for the parametric test because the number of rooms did not meet the requirement of sample size for a parametric test and the data was not normally distributed, therefore nonparametric test was used. Wilcoxon signed-rank tests were used to calculate the mean differences in room hygiene before and after the training, and a Kruskal-Wallis test was used to assess differences between interventions.

## Training intervention

A hygiene training intervention was developed to improve both the personal and room hygiene of the santris. Training was delivered to santris by the supervisors, and training for the trainers was conducted by the first author. The duration of the training was approximately 4 hours, and the training for the trainers was 8 hours. The training materials taught supervisors how to assess personal hygiene and room hygiene, deliver the training intervention using the materials provided (video, poster, and leaflets), and maintain personal and room hygiene, and they led santris in a discussion about the importance of personal and room hygiene.

For each mode of intervention, a presentation by the supervisor was given prior to using the video, poster, and/or leaflet, and ended with a discussion. Santris in this Islamic boarding school are still teenagers (11–13 years). At that age, interventions need to be 3S (simple, scalable, and sustainable) [22]. The sequence of the intervention was not randomized, i.e., a video was shown for the first cluster, followed by poster for the second cluster, and leaflets for individual santris in the last cluster. The order of intervention methods was designed to reduce the likelihood of introducing bias among subsequent clusters that used different intervention methods. The time interval between clusters was one week. At the end of each training method, all santris were asked to clean their room. The duration of the training ranged between 180 and 240 minutes. The content of the intervention was similar, covering hygiene principles from the Islamic perspective (e.g., level of purification, type of *najis* and how to purify it), hygiene with the goal of disease prevention, personal hygiene, and room hygiene. The personal hygiene materials emphasized head-to-toe daily personal hygiene, such as taking a bath, brushing the teeth, washing the hands, changing underwear, and using clean clothes. Room hygiene materials contained ways to maintain room cleanliness, including cleaning the floor, making the bed, and tidying rubbish from the area. In total, the intervention was eight weeks long (Fig 2).

## Ethics approval and consent to participate

Ethical approval for this study was obtained from the Medical and Health Research Ethics Committee (MHREC), Faculty of Medicine, Public Health and Nursing, Universitas Gadjah Mada (Ref: KE/FK/725/EC/2016). Informed consent was received from both parents and the santris in the boarding school.

## Results

### Qualitative

The FGD participants' educational backgrounds were largely senior high school level (19 persons), followed by undergraduate and graduate levels (one person each). Most FGD participants and in-depth interviewees were female (15 of a total 21). Only one informant (female, older than 20 years) had a master's level education.

Based on the FGDs and in-depth interviews, we identified santris' poor personal and room hygiene practices, as well as potential solutions that supervisors proposed. Poor practices were

| | Week 1 | Week 2 | Week 3 | Week 4 | Week 5 | Week 6-8 |
|---|---|---|---|---|---|---|
| *Cluster* 1: Video | Pretest personal and room hygiene | Knowledge pretest, A-H girls room and A-F boys room intervention | Posttest knowledge, personal and room hygiene | Posttest personal and room hygiene | Posttest personal and room hygiene | Posttest personal and room hygiene |
| *Cluster* 2: Poster | Pretest personal and room hygiene | Pretest personal and room hygiene | Knowledge pretest, I-P girls room and G-M boys room intervention | Posttest knowledge, personal and room hygiene | Posttest personal and room hygiene | Posttest personal and room hygiene |
| *Cluster* 3: Leaflet | Pretest personal and room hygiene | Pretest personal and room hygiene | Pretest personal and room hygiene | Knowledge pretest, Q-X girls room and N-T boys room intervention | Posttest knowledge, personal and room hygiene | Posttest personal and room hygiene |

**Fig 2. Implementation of research intervention in the Islamic boarding school.**

possibly related to internal factors, such as the santris' relatively young age, lack of independence in managing themselves, low awareness of cleanliness and hygiene practices, laziness, stubbornness, and low sense of belonging to the provided facilities. In addition, external pressures contributed to poor hygiene practices, including santris' schedule of activities, influence of roommates and their willingness to maintain a shared space, and parents' provision of living items. Family socio-economic may have also impacted behavior.

Despite existing policies and programs for encouraging hygiene practices, supervisors acknowledged that problems remained. Therefore, several potential solutions were proposed, including greater control of the santris' behavior, strengthening regulations, communication with parents, and hygiene training. Hygiene training was perceived as an entry point prior to greater control and regulation. Therefore, training was selected as the intervention for this study to cover topics such understanding hygiene from the Islamic perspective and maintaining personal and room hygiene (Table 1).

## Quantitative

The total number of first grade junior high school santris was 512. Among these santris, 452 (88.3%) completed the study (258 females and 194 males; Fig 1). Fifty-six santris did not return

**Table 1. Quotations related to hygiene problems and potential solutions.**

| Topics | Quotations |
|---|---|
| **Santris' hygiene condition** | |
| Hygiene condition | *"Sometimes [they] dispose of their garbage carelessly and that is contagious."(II SP)* |
| | *"There is one student in the first year who does not take a bath for a week. He rarely takes a bath or changes clothes."(FGDMale, I6)* |
| Hygiene expectation | *"A clean room. The parameter is they clean the room after having meals. Dirty clothes are put together and are separated from the clean ones, on the chart."(FGDFemale, I5)* |
| Policy | *"We form the officers and its structure, who are the coordinator, controller, and the supervisors"(II SP)* |
| Reasoning | *"Age, less awareness" (FGDFemale, I1)* |
| | *"Maybe they don't know how to do it or basically they don't care" (FGDFemale, I9)* |
| | *"His parents send clothes almost every day, dress changes every day. . . Dirty clothes are fetched by his friends, they are not used anymore." (FGDFemale, I7)* |
| **Supervisors potential solution** | |
| Existing programs | *"For cleanliness, e.g., room competition for cleanliness" (FGDFemale, I4)* |
| | *"Usually, the program is running well in the beginning and then disappears after a while!" (FGDMale, I4)* |
| Alternative solution | *"First, do hygiene promotion first, and then they are given a reward. After a while, they need to be independent, supported by someone functioning as a supervisor to control the behavior and decide the situation when the children should be given straightforward instruction" (II SP)* |
| Training material | *"The materials should be motivating. There should be role models who are already successful, simple things to do, and fun. Another example is the benefit of hygiene, from the religion point of view and its application. They also need to understand what is the impact to their health in the long term." (II SP)* |

the consent form because they could not meet with their parents, they lost the consent form, or were absent from school.

Overall, the participants' reaction toward the training was positive, as indicated by the average score greater than 4 out of a maximum of 5 (Table 2). The highest score was given for the benefit of the training among both female and male participants. Males reacted more positively than females, and differences were significant for all areas except for training materials.

A significant increase in knowledge among both boys and girls across all intervention types was found in the scores before and after training (7.22 ± 1.34 pretraining and 7.70 ± 0.74 post training; $p$-value <0.001). Improvements over the study period were larger for males and those who received posters. However, the post training knowledge levels were higher among females and those who watched the video (Table 3). A Bonferroni post hoc test revealed that

**Table 2. Reaction evaluation toward hygiene training by gender.**

| Variables | Boys | Girls | Total | $p$-value* |
|---|---|---|---|---|
| | Mean ± SD (95% CI) | Mean ± SD (95% CI) | Mean ± SD (95% CI) | |
| Benefits of training | 4.59 ± 0.63 (4.50–4.68) | 4.72 ± 0.57 (4.65–4.79) | 4.66 ± 0.60 (4.60–4.71) | 0.015 |
| Training materials | 4.34 ± 0.71 (4.24–4.44) | 4.34 ± 0.64 (4.26–4.42) | 4.34 ± 0.67 (4.28–4.40) | 0.438 |
| Training process | 4.24 ± 0.82 (4.12–4.36) | 4.10 ± 0.67 (4.02–4.18) | 4.16 ± 0.74 (4.09–4.23) | 0.022 |
| Satisfaction with the training management process | 4.43 ± 0.75 (4.32–4.54) | 4.24 ± 0.70 (4.15–4.32) | 4.32 ± 0.72 (4.25–3.87) | 0.003 |
| Satisfaction with the coach | 4.48 ± 0.79 (4.37–4.59) | 4.26 ± 0.71 (4.17–4.34) | 4.36 ± 0.75 (4.29–4.43) | 0.001 |

* Independent T-test

**Table 3. Knowledge and behavior evaluation before and after hygiene training by gender and intervention method.**

| Groups | Pretest | Posttest | Paired T-test | Between groups |
|--------|---------|----------|---------------|----------------|
| | Mean ± SD (95%CI) | Mean ± SD (95%CI) | | |
| **Knowledge** | | | | |
| All groups | 7.22 ± 1.34 (7.10–7.34) | 7.70 ± 0.74 (7.63–7.77) | < 0.001 | |
| Girls | 7.40 ± 0.95 (7.28–7.52) | 7.74 ± 0.72 (7.65–7.82) | < 0.001 | < 0.001* |
| Boys | 6.98 ± 1.70 (6.73–7.22) | 7.65 ± 0.77 (7.54–7.76) | < 0.001 | |
| Video groups | 7.35 ± 1.01 (7.18–7.34) | 7.83 ± 0.53 (7.75–7.92) | < 0.001 | 0.025** |
| Poster groups | 7.01 ± 1.75 (6.74–7.28) | 7.65 ± 0.83 (7.52–7.77) | < 0.001 | |
| Leaflet groups | 7.34 ± 1.02 (7.16–7.50) | 7.62 ± 0.79 (7.49–7.53) | < 0.001 | |
| **Behavior** | | | | |
| All groups | 9.75 ± 2.98 (9.47–10.03) | 12.16 ± 2.12 (11.97–12.36) | <0.001 | |
| Girls | 10.14 ± 2.42 (9.84–10.44) | 11.74 ± 2.12 (11.49–12.00) | <0.001 | 0.067* |
| Boys | 9.29 ± 3.52 (8.74–9.74) | 12.72 ± 1.99 (12.44–13.01) | <0.001 | |
| Video groups | 7.73 ± 3.21 (7.20–8.26) | 11.76 ± 2.10 (11.41–12.11) | <0.001 | <0.001** |
| Poster groups | 10.89 ± 2.46 (10.52–11.27) | 12.95 ± 1.83 (12.67–13.23) | <0.001 | |
| Leaflet groups | 10.42 ± 2.19 (10.06–11.79) | 11.64 ± 2.18 (11.23–12.00) | <0.001 | |

*Independent T-Test

** One-way ANOVA

video intervention was significantly higher than leaflet group (*p*-value = 0.041). There was no statistical significant difference between video intervention compared to poster group (*p*-value = 0.070) and poster intervention compared to leaflet intervention (*p*-value = 1.000).

A similar finding was observed with respect to changes in personal hygiene over the course of the intervention (9.75 ± 2.98 pretraining and 12.16 ± 2.12 posttraining; *p*-value < 0.001). In comparison to females and participants receiving other intervention methods, males and those who watched the video experienced the largest improvements, with the highest scores observed among males and those who received poster intervention (Table 3). A Bonferroni post hoc test revealed that poster intervention was significantly higher than video intervention (*p*-value < 0.001) and leaflet intervention (*p*-value < 0.001). There was no statistical significant difference between video intervention compared to leaflet intervention (*p*-value = 1.000).

The increase in knowledge and personal hygiene behavior, however, did not appear to have a significant effect on room hygiene. Overall, room hygiene increased, but the changes were not statistically significant. Significant differences were only observed among males and those who received the leaflet intervention (Table 4). The improvement in room hygiene was greater among males than females, unlike that for personal hygiene. The improvements in knowledge and change of behavior in personal hygiene observed among males were also higher than those observed among females (0.67 vs. 0.34 and 3.43 vs. 1.6, respectively).

## Discussion

This study emphasized the important role school-based interventions can play in improving students' hygiene practices. The study was designed to test the effect of hygiene training interventions on hygiene practices in an Islamic boarding school setting.

Our findings demonstrated that hygiene training significantly improves knowledge and personal hygiene behavior among the santris. Unlike personal hygiene, improvement in knowledge, however, may not have practical implications since knowledge levels were already high prior to the intervention. This could reflect previous efforts undertaken within the

**Table 4. Room hygiene before and after hygiene training by gender and intervention method.**

| Groups | Pretest | Posttest | Wilcoxon Rank Test | Between groups |
|---|---|---|---|---|
| | Median (IQR) | Median (IQR) | | |
| All groups | 5.00 (3.67) | 5.50 (4.23) | 0.140 | |
| Girls | 4.83 (4.08) | 4.20 (2.47) | 0.548 | <0.001* |
| Boys | 5.58 (3.00) | 7.20 (2.92) | 0.001 | |
| Video groups | 5.00 (5.00) | 4.93 (2.28) | 0.245 | 0.812** |
| Poster groups | 5.50 (5.00) | 5.83 (2.67) | 0.477 | |
| Leaflet groups | 5.00 (3.00) | 5.95 (4.60) | 0.017 | |

*Mann-Whitney

** Kruskal-Wallis

school's program. Other studies have also shown that several training methods have been proven to significantly improve hygiene. These methods include a program of interactive presentations on toilet hygiene for 11- to 16-year-old Turkish schoolchildren [23]. Interactive interventions with proven effectiveness included demonstrations involving children, posters [24], cartoon or manga characters [25–27], videos, and leaflets [28].

Regarding the personal hygiene behavior analyses, despite significant improvements after the training, work is still needed to improve behaviors, because less than half of the santris practiced poor personal hygiene. These behaviors included handwashing with soap for at least twenty seconds before and after meals (27% at baseline vs. 33% after training), washing utensils immediately after use (36% vs. 49% after training) and changing bed sheets weekly (45% vs. 54% after training). Limited improvement was observed with respect to handwashing, washing utensils, and changing bed sheets, as these behaviors were strongly related to access to Water, Sanitation and Hygiene (WASH) facilities. Even though knowledge, attitude, and practice were adequate, the lack of resources such as soap may seriously impede hygiene practices [29].

With respect to room hygiene, improvements were only significant among male santris and those who received leaflet, further qualitative observation identified several factors that could explain why males practiced better room hygiene than females (Table 4). Males appeared to have fewer items in their rooms, their supervisors tended to be more attentive toward hygiene in males' rooms, and activities outside the classroom commonly took place in the mosque or classroom instead of their bedrooms. Females santris tended to have more items of clothing, bed linen, and cosmetics. In addition to being more critical of their female supervisors' own hygiene habits and behaviors, female santris also tended to spend leisure time in their rooms, such as eating, chatting and studying. James' study on adolescent girls found a similar evidence, with girls felt comfortable doing most activities in their room, including both physical and non-physical activities [30].

Junior high school santris in this boarding school are familiar with use of video, poster or leaflet as intervention medias. All forms of media used in this study significantly improved knowledge and personal behavior. However, only leaflets significantly improved room hygiene. This finding may be due to the advantage of leaflets as opposed to other medias used (i.e. poster and video) or timing of intervention. As in other studies, leaflets appeared to be most promising, because they can be distributed individually, are portable, and can be read many times, making this medium effective for health education among teenagers [31,32]. In addition, with leaflet as the last intervention, sharing of information among the santris may accumulate with time factor which cannot be controlled under the study design applied. Posters were most effective with respect to personal hygiene, while videos led to the largest

improvements in knowledge. Video, leaflet, and poster intervention methods were also found to be effective in educating primary and secondary students in studies conducted in Northern Tanzania [32] and targeted to youth aged ten to 18 years [33]. Therefore, all media can be used in interventions, and media can be selected based on predefined training goals.

This study had limitations due to only one site study with non-random allocation of participants into the cluster intervention groups, as reflected by the lower pretest scores of knowledge in the poster intervention group and lower pretest scores of personal hygiene among those in the video intervention group. Regarding the possible threat of diffusion, we were not able to fully minimize the possibilities of occurring in this study, due to arrangements in the santris' daily activities and sleeping arrangement in this congregate boarding school setting. To reduce the risk of bias, we first applied the interventions using video, followed by posters and leaflets. The santris only watched the video during the intervention and did not have access to the files. While for the posters, they were only displayed in the santris' rooms who belong to this intervention cluster. Leaflet was implemented as the last intervention because of the possibilities for the santris' to share with others. For future research, we suggest to minimize this threat using a more rigorous study design such as the counterbalanced design.

## Conclusion

This study concludes that hygiene practices are still problematic in congregate settings, such as Islamic boarding school. Even though training intervention has been specifically designed from the Islamic perspective based on findings from the qualitative study and santris were satisfied with the intervention, school-based hygiene training was effective to improve knowledge about the importance of hygiene and personal behavior. However, to strengthen its effect on room hygiene, particularly among the females, modifications of interventions and a comprehensive approach are necessary.

## Supporting information

**S1 Appendix.**
(DOCX)

## Acknowledgments

We thank the boarding school and all participants (particularly the supervisors and santris) for their involvement in this study. This study was conducted as a partial requirement to fulfill a Master of Public Health Degree in International Health in the Graduate Program of Public Health, Faculty of Medicine, Public Health and Nursing, Universitas Gadjah Mada, Yogyakarta, Indonesia.

## Author Contributions

**Conceptualization:** Vita Widyasari, Yayi Suryo Prabandari, Adi Utarini.

**Data curation:** Vita Widyasari.

**Formal analysis:** Vita Widyasari, Yayi Suryo Prabandari, Adi Utarini.

**Investigation:** Vita Widyasari.

**Methodology:** Vita Widyasari, Yayi Suryo Prabandari, Adi Utarini.

**Project administration:** Vita Widyasari.

**Writing – original draft:** Vita Widyasari, Yayi Suryo Prabandari, Adi Utarini.

**Writing – review & editing:** Vita Widyasari, Yayi Suryo Prabandari, Adi Utarini.

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
