## [Decision Letter · Decision Letter 0]

28 Oct 2019

PONE-D-19-26714

Training intervention to improve hygiene practices in Islamic Boarding Schools in Yogyakarta, Indonesia: A mixed methods study

PLOS ONE

Dear Prof Widyasari,

Thank you for submitting your manuscript to PLOS ONE. After careful consideration, we feel that it has merit but does not fully meet PLOS ONE’s publication criteria as it currently stands. Therefore, we invite you to submit a revised version of the manuscript that addresses the points raised during the review process.

You will see from the comments that two reviewers recommended rejecting the article. However, I would like to see if you can address the points of concern. Please carefully consider all 4 reviewer comments and respond accordingly. 

We would appreciate receiving your revised manuscript by 26 December 2019. To enhance the reproducibility of your results, we recommend that if applicable you deposit your laboratory protocols in protocols.io, where a protocol can be assigned its own identifier (DOI) such that it can be cited independently in the future. For instructions see: http://journals.plos.org/plosone/s/submission-guidelines#loc-laboratory-protocols

We look forward to receiving your revised manuscript.

Kind regards,

Andrew Soundy

Academic Editor

PLOS ONE

Journal Requirements:

1. 

2.  Please refer to any post-hoc corrections to correct for multiple comparisons during your statistical analyses. if these were not performed please justify the reasons. Additionally, in the text you state: "The subjects were nonrandomly assigned to one of three different interventions". Please discuss how this assigning occurred.

3.  Please provide further details concerning the piloting of the interview guide used in this study. I.e. who were the participants and how many were they.

4.  Please remove your figures from within your manuscript file, leaving only the individual TIFF/EPS image files, uploaded separately.  These will be automatically included in the reviewers’ PDF.

Reviewers' comments:

Reviewer's Responses to Questions

**Comments to the Author**

1. Is the manuscript technically sound, and do the data support the conclusions?

Reviewer #1: Yes

Reviewer #2: Yes

Reviewer #3: No

Reviewer #4: Partly

2. Has the statistical analysis been performed appropriately and rigorously? 

Reviewer #1: Yes

Reviewer #2: Yes

Reviewer #3: Yes

Reviewer #4: No

3. Have the authors made all data underlying the findings in their manuscript fully available?

Reviewer #1: Yes

Reviewer #2: Yes

Reviewer #3: Yes

Reviewer #4: Yes

4. Is the manuscript presented in an intelligible fashion and written in standard English?

Reviewer #1: Yes

Reviewer #2: No

Reviewer #3: No

Reviewer #4: Yes

5. Review Comments to the Author

Reviewer #1: The article is on the efficacy of different training intervention on promoting hygiene behaviours among boarding students. I have attached comments and suggestions to help improve the article. The comments are made in line with the different sections of the article

Reviewer #2: Dear Authors

The manuscript entitled "TRAINING INTERVENTION TO IMPROVE HYGIENE PRACTICES IN ISLAMIC BOARDING SCHOOLS IN YOGYAKARTA, INDONESIA: A MIXED METHODS STUDY" is very important issues of health and hygiene in school health areas.

Please see our comments for this manuscript on the list below which attached in this letter.

Thank you.

Sincerely yours,

Reviewer

Reviewer #3: The experiment have not been conducted rigorously. Since the intervention was not conducted at the same time, it was really possible student at the second and third phase of intervention have been exposed before intervention by student from the first phase. The author also didn't mentioned how was the sleeping arrangement among those student. If any student from the first phase intervention share the same room with student from the second or third intervention, they might be shared any information. Furthermore, the study found that the leaflet group has a significant change on room hygiene. Since the group was intervened lastly, the evaluation period was shorter than other groups.

Reviewer #4: COMMENTS TO AUTHORS

Let me congratulate authors for taking time to research into such an important phenomenon. I find the following comments useful for enriching the content of the manuscript.

Abstract

“The primary objective of this study was to measure the impact of a training intervention on improving hygiene practices among Islamic boarding school students…”.

I caution authors against the use of the word “impact”. It is difficult for authors to convince readers that they truly measured impact. On the measurement chain, impact is in the very long term sometimes 10 years and more. This cannot be practicable for a hand hygiene study of this nature.

Introduction

In a 2014-2015 school health report, 70% of santris in a boarding school 92 were diagnosed with infectious diseases, predominantly respiratory and skin infections..”

Cite the source document appropriately.

Materials and Methods

Study Design

This section needs some clarity. First of all;

i. what constituted your clusters?

ii. How did you control for clustering in considering a sample size?

iii. What was the unit of analysis?

iv. How did you decide on the number of students to participate in the study? For eg, did you estimate the figure statistically?

Data Collection and analysis

Qualitative study

i. How did you derive the 21 supervisors?

ii. What is the source of your data collection tools?

iii.

Quantitative study

i. What was the source of quantitative data collection tool?

ii. You mentioned the use of a Likert scale. How did you test for the internal reliability of items assessing the same variable?

iii. How did you control for clustering at the data analysis stage?

Training intervention

i. How did you ensure that training was the same for the different intervention groups?

ii. Did you assess the intervention fidelity to ensure that the interventions delivered were in adherence to the protocols of the study?

Results

It appears you statistically compared the outcomes within the boys groups with the outcomes within the girls group at some point. Such a comparison will require unpaired samples t-test and not a paired samples t-test.

6. PLOS authors have the option to publish the peer review history of their article (what does this mean?). If published, this will include your full peer review and any attached files.

Reviewer #1: No

Reviewer #2: No

Reviewer #3: No

Reviewer #4: No

---

## [Author Response · Author response to Decision Letter 0]

3 Jan 2020

We have responded to all comments from the academic editor and three reviewers in the attaches file.

---

## [Decision Letter · Decision Letter 1]

30 Jan 2020

PONE-D-19-26714R1

Training intervention to improve hygiene practices in Islamic boarding school in Yogyakarta, Indonesia: a mixed-method study

PLOS ONE

Dear Dr Widyasari,

Thank you for submitting your manuscript to PLOS ONE. After careful consideration, we feel that it has merit but does not fully meet PLOS ONE’s publication criteria as it currently stands. Therefore, we invite you to submit a revised version of the manuscript that addresses the points raised during the review process.

We would appreciate receiving your revised manuscript by 30 February 2020. To enhance the reproducibility of your results, we recommend that if applicable you deposit your laboratory protocols in protocols.io, where a protocol can be assigned its own identifier (DOI) such that it can be cited independently in the future. For instructions see: http://journals.plos.org/plosone/s/submission-guidelines#loc-laboratory-protocols

We look forward to receiving your revised manuscript.

Kind regards,

Andrew Soundy

Academic Editor

PLOS ONE

Additional Editor Comments (if provided):

Please consider the points made by the reviewers wanting changes.

Reviewers' comments:

Reviewer's Responses to Questions

**Comments to the Author**

1. If the authors have adequately addressed your comments raised in a previous round of review and you feel that this manuscript is now acceptable for publication, you may indicate that here to bypass the “Comments to the Author” section, enter your conflict of interest statement in the “Confidential to Editor” section, and submit your "Accept" recommendation.

Reviewer #1: All comments have been addressed

Reviewer #2: All comments have been addressed

Reviewer #4: (No Response)

2. Is the manuscript technically sound, and do the data support the conclusions?

Reviewer #1: Yes

Reviewer #2: Partly

Reviewer #4: Partly

3. Has the statistical analysis been performed appropriately and rigorously? 

Reviewer #1: Yes

Reviewer #2: Yes

Reviewer #4: No

4. Have the authors made all data underlying the findings in their manuscript fully available?

Reviewer #1: Yes

Reviewer #2: Yes

Reviewer #4: Yes

5. Is the manuscript presented in an intelligible fashion and written in standard English?

Reviewer #1: Yes

Reviewer #2: No

Reviewer #4: No

6. Review Comments to the Author

Reviewer #1: I have carefully read the author's responses to the reviewers and I am satisfied they addressed them accordingly. They heeded to the reviewers suggestions. A bit of article clean-up will help now, which can be handled through the editorial.

Reviewer #2: Dear Authors,

Thank you for re-submit your revision manuscript. However, I didn't found your comments from the reviewers. Please answer one by one the comments of reviewer. You didn't answered the reviewer comments. Therefore, it is made reviewer confusing. Please make a highlight with yellow colour in part of your manuscript where placed the authors revision.

If the authors made it, it is more easier the reviewer for review.

Thank you.

Regards,

Reviewer

Reviewer #4: I do not see any of my earlier comments addressed by the authors.

COMMENTS TO AUTHORS

Abstract

“The primary objective of this study was to measure the impact of a training intervention on improving hygiene practices among Islamic boarding school students…”.

Once again, I find it necessary for authors to replace the word “impact”. It is difficult for authors to convince readers that they truly measured impact. On the measurement chain, impact is in the very long term sometimes 10 years and more. This cannot be practicable for a hand hygiene study of this nature. They may consider outcome or effect instead.

Introduction

In a 2014-2015 school health report, 70% of santris in a boarding school 92 were diagnosed with infectious diseases, predominantly respiratory and skin infections..”

Cite the source document appropriately.

Materials and Methods

Study Design

This section needs some clarity. First of all;

i. what constituted your clusters?

ii. How did you control for clustering in considering a sample size?

iii. What was the unit of analysis?

iv. How did you decide on the number of students to participate in the study? For eg, did you estimate the figure statistically?

Data Collection and analysis

Qualitative study

i. How did you derive the 21 supervisors?

ii. What is the source of your data collection tools?

iii.

Quantitative study

i. What was the source of quantitative data collection tool?

ii. You mentioned the use of a Likert scale. How did you test for the internal reliability of items assessing the same variable?

iii. How did you control for clustering at the data analysis stage?

Training intervention

i. How did you ensure that training was the same for the different intervention groups?

ii. Did you assess the intervention fidelity to ensure that the interventions delivered were in adherence to the protocols of the study?

Results

It appears you statistically compared the outcomes within the boys groups with the outcomes within the girls group at some point. Such a comparison will require unpaired samples t-test and not a paired samples t-test.

7. PLOS authors have the option to publish the peer review history of their article (what does this mean?). If published, this will include your full peer review and any attached files.

Reviewer #1: No

Reviewer #2: No

Reviewer #4: No

---

## [Author Response · Author response to Decision Letter 1]

26 Feb 2020

Thank you for the opportunity to revise our manuscript, entitled “Training intervention to improve hygiene practices in Islamic boarding schools in Yogyakarta, Indonesia: a mixed methods study”. We appreciate the constructive suggestions given by the reviewers and we have made our best attempts to provide responses to the reviewers. 

Please find our responses to the editor and four reviewer comments in blue colour font in the attachment files.

We sincerely hope that you will find our manuscript suitable for publication and look forward to hearing the final decision from you in due course.

---

## [Decision Letter · Decision Letter 2]

18 Mar 2020

PONE-D-19-26714R2

Training intervention to improve hygiene practices in Islamic boarding school in Yogyakarta, Indonesia: a mixed-method study

PLOS ONE

Dear Prof Widyasari,

Thank you for submitting your manuscript to PLOS ONE. After careful consideration, we feel that it has merit but does not fully meet PLOS ONE’s publication criteria as it currently stands. Therefore, we invite you to submit a revised version of the manuscript that addresses the points raised during the review process.

See comments below. 

We would appreciate receiving your revised manuscript by 18 April 2020. To enhance the reproducibility of your results, we recommend that if applicable you deposit your laboratory protocols in protocols.io, where a protocol can be assigned its own identifier (DOI) such that it can be cited independently in the future. For instructions see: http://journals.plos.org/plosone/s/submission-guidelines#loc-laboratory-protocols

We look forward to receiving your revised manuscript.

Kind regards,

Andrew Soundy

Academic Editor

PLOS ONE

Additional Editor Comments (if provided):

Please make the minor revisions requested. Once this is achieved the paper will be accepted.

Reviewers' comments:

Reviewer's Responses to Questions

**Comments to the Author**

1. If the authors have adequately addressed your comments raised in a previous round of review and you feel that this manuscript is now acceptable for publication, you may indicate that here to bypass the “Comments to the Author” section, enter your conflict of interest statement in the “Confidential to Editor” section, and submit your "Accept" recommendation.

Reviewer #2: All comments have been addressed

Reviewer #4: All comments have been addressed

2. Is the manuscript technically sound, and do the data support the conclusions?

Reviewer #2: Yes

Reviewer #4: Yes

3. Has the statistical analysis been performed appropriately and rigorously? 

Reviewer #2: Yes

Reviewer #4: Yes

4. Have the authors made all data underlying the findings in their manuscript fully available?

Reviewer #2: Yes

Reviewer #4: Yes

5. Is the manuscript presented in an intelligible fashion and written in standard English?

Reviewer #2: Yes

Reviewer #4: Yes

6. Review Comments to the Author

Reviewer #2: Comments:

1. In Introduction Line 78-83, please add references the authors cites?

2. Line 90-92, why the authors selected mixed method? What they reasons? Regarding previous studies in Indonesian setting.

3. In method, because just one sites study, the authors should explain their limitation for generalization?

4. In qualitative study, please explain the saturation of data? How the researchers stop their interview for saturation data?

5. In FGD, are the authors conducted separately the FGD? How they divided the group? Related to saturation data? Any data observation?

6. Please add your limitation of study?

Reviewer #4: An intervention research on the subject of hand hygiene is crucial especially at this time of a pandemic due to infectious disease. The authors have addressed my comments satisfactorily.

7. PLOS authors have the option to publish the peer review history of their article (what does this mean?). If published, this will include your full peer review and any attached files.

Reviewer #2: No

Reviewer #4: No

---

## [Author Response · Author response to Decision Letter 2]

14 Apr 2020

Thank you for the opportunity to revise our manuscript, entitled “Training intervention to

improve hygiene practices in Islamic boarding schools in Yogyakarta, Indonesia: a

mixed methods study”. We appreciate the constructive suggestions given by the

reviewers and we have made our best attempts to provide responses to the reviewers.

Please find our responses to reviewer 2 comments in blue colour font

in the attachment files.

We sincerely hope that you will find our manuscript suitable for publication and look

forward to hearing the final decision from you in due course.

---

## [Decision Letter · Decision Letter 3]

4 May 2020

Training intervention to improve hygiene practices in Islamic boarding school in Yogyakarta, Indonesia: a mixed-method study

PONE-D-19-26714R3

Dear Dr. Widyasari,

We are pleased to inform you that your manuscript has been judged scientifically suitable for publication and will be formally accepted for publication once it complies with all outstanding technical requirements.

With kind regards,

Andrew Soundy

Academic Editor

PLOS ONE

Additional Editor Comments (optional):

Reviewers' comments:

Reviewer's Responses to Questions

**Comments to the Author**

1. If the authors have adequately addressed your comments raised in a previous round of review and you feel that this manuscript is now acceptable for publication, you may indicate that here to bypass the “Comments to the Author” section, enter your conflict of interest statement in the “Confidential to Editor” section, and submit your "Accept" recommendation.

Reviewer #2: All comments have been addressed

2. Is the manuscript technically sound, and do the data support the conclusions?

Reviewer #2: Yes

3. Has the statistical analysis been performed appropriately and rigorously? 

Reviewer #2: Yes

4. Have the authors made all data underlying the findings in their manuscript fully available?

Reviewer #2: Yes

5. Is the manuscript presented in an intelligible fashion and written in standard English?

Reviewer #2: Yes

6. Review Comments to the Author

Reviewer #2: I studied revised manuscript, Thank you for the opportunity to review this manuscript. in my opinion, the authors have made all the required corrections and the article is acceptable.

7. PLOS authors have the option to publish the peer review history of their article (what does this mean?). If published, this will include your full peer review and any attached files.

Reviewer #2: Yes: Tantut Susanto

---

## [Editor Report · Acceptance letter]

8 May 2020

PONE-D-19-26714R3 

Training intervention to improve hygiene practices in Islamic boarding school in Yogyakarta, Indonesia: a mixed-method study 

Dear Dr. Widyasari:

I am pleased to inform you that your manuscript has been deemed suitable for publication in PLOS ONE. Congratulations! Your manuscript is now with our production department. 

With kind regards,

on behalf of

Dr. Andrew Soundy 

Academic Editor

PLOS ONE